

**Why Is Height-Dependent Mixing Observed in Stratocumulus?**
Zeen Zhu[1], Fan Yang[1], Steven Krueger[2], Yangang Liu[1]
[1]Environmental Science and Technologies Department, Brookhaven National Laboratory, Upton,
NY, 11973, USA
[2]Department of Atmospheric Science, University of Utah, Salt Lake City, UT, 84112, USA
*Correspondence to*: Zeen Zhu (zzhu1@bnl.gov)
**Abstract.** Recent aircraft measurements in stratocumulus clouds suggest that entrainment mixing
is inhomogeneous (IM) near cloud top and homogeneous (HM) within the cloud. However, this
proposed height-dependence of mixing transition is uncertain because of artifacts involved in the
aircraft measurements. In this study, we use the Explicit Mixing Parcel Model to simulate mixing
scenarios in stratocumulus clouds and reconstruct the virtual aircraft measurements to investigate
the mixing signature. Results show that, from the aircraft-measurement perspective, the mixing
signature always exhibits IH characteristic near cloud top and HM characteristic within cloud,
independent of the types of the local entrainment-mixing process. The appearance of the vertical
IM-to-HM transition is essentially a collective behavior of multiple parcels sampled at the same
height, experiencing distinct entrainment-mixing-evaporation histories. This bulk view of mixing
process, which is widely used for aircraft measurements, could lead to misinterpretations of the
true mixing mechanism occurring in clouds. Our result underscores the limitations of using aircraft
measurements to identify the local entrainment-mixing mechanism at the process level.








## 1. Introduction

Entrainment-mixing is a critical cloud process and plays important roles in simulating precipitation
formation, radiative properties and macroscopic structures (Lasher-Trapp et al., 2005; Baker et al.,
1980; Lehmann et al., 2009; Magaritz-Ronen et al., 2014; Chosson et al., 2007). In the
stratocumulus, entrainment-mixing is initiated near cloud top where the dry, warm free-
troposphere air is partially mixed with the cloudy air and then entrained (Wood, 2012). After
entrainment, cloud droplets start to evaporate in a subsaturated environment along with the mixing
process. Depending on the efficiency of mixing and evaporation, two mixing scenarios are
generally considered: homogeneous mixing (HM) and inhomogeneous mixing (IM) (Latham and
Reed, 1977; Baker et al., 1980). For HM, turbulent mixing is much faster than droplet evaporation.
Under the extreme condition, the cloudy air is mixed immediately with the entrained air such that
all cloud droplets are exposed to the same sub-saturation environment, resulting in reduced droplet
size and unchanged number concentration. For the IM, turbulent mixing is slower than evaporation.
Under the IM condition, cloud droplets adjacent to the dry entrained air are quickly evaporated
while leaving the remaining droplets unaffected.
Over the recent decades, a consensus has emerged from aircraft observations across multiple field
campaigns that stratocumulus clouds tend to exhibit IM signature near the cloud top and a HM
signature in the mid-levels (Yum et al., 2015; Yeom et al., 2021; Desai et al., 2021; Wang et al.,
2009; Gao et al., 2021). One hypothesis to explain this behavior is the "vertical circulation"
concept which is proposed by Wang et al. (2009), further refined by Yum et al. (2015) and detailed
in Yeom et al. (2021). Specifically, after entrainment occurs near cloud top, the cloud parcel starts
to descend. The droplets in the diluted descending parcels evaporate and reduce the particle sizes.
Therefore, if the mid-level cloud is horizontally sampled by the aircraft, droplets are likely to be
evaporated in the diluted regions than those in the undiluted regions, leading to the HM signature
in the middle of cloud. Yeom et al. (2023) further conducted experiments in the cloud chamber by
injecting dry air into the well-mixed cloud to mimic the entrainment–mixing process. Result shows
that cloud microphysical responses to entrainment and mixing are locally inhomogeneous and
globally homogeneous, implying that the global versus local sampling of clouds can lead to
contradictory mixing results. These studies provide critical insights to reevaluate the applicability
of using aircraft measurements for HM/IM mixing classification.
Conventionally, cloud microphysical properties (e.g., droplet number and size) measured by
aircraft flying along a horizontal path are used to calculate the mixing metrics (see section 2.2) for
IM/HM classification. However, this aircraft-based perspective is known with several issues: 1):
the global mean cloud properties are not representative of the cloud structures at small scales. For
instance, Allwayin et al. (2024) utilizes holographic measurements showing that droplet size
distributions are more narrow at small scales than those at whole-cloud averages. 2) If the mixing
in each small sampling is inhomogeneous, then an average of several samplings may lead to



apparent homogeneous mixing (Burnet and Brenguier, 2007); 3) the aircraft flying along a path at
the same height measures a collection of mixed air parcels with different entrainment-mixing
stages, this collected behavior from various mixing parcels may not represent the original mixing
process in each individual parcel (Yeom et al., 2023). In this study, we revisit the applicability of
using aircraft measurements for mixing identification. We design a simulation framework based
on the Explicit Mixing Parcel Model (EMPM) to emulate the aircraft measurements in the Sc. We
show that, using aircraft measurements, the mixing behavior in Sc is always identified as IH near
cloud top and HM within clouds, regardless of the local mixing scenario within individual parcels.
The layout of this paper is organized as follows: Section 2 introduces the EMPM model, including
the adapted assumptions and the experiment set up. The mixing metrics used for HM/IM
identification applied in this study are introduced. In Section 3, the EMPM simulations are
analyzed from two perspectives: bulk and local. We show that, based on the same simulation
output, the mixing process in clouds may exhibit differently from the two perspectives; this
discrepancy is the key to understanding the limitations of aircraft measurements. In section 3, we
also conducted an additional isobaric mixing experiment to isolate the mixing and adiabatic
warming process which are coexisting in previous experiments. In Section 4, we explain the
phenomenon of the IM-HM transition in Sc and discuss the insights on future mixing studies.
Finally, a conclusion constitutes Section 5.

## 2. Methods

### 2.1 Experiment Design

The Explicit Mixing Parcel Model (EMPM) was developed by Krueger et al. (1997) to simulate
the evolution of cloud thermodynamic properties influenced by turbulent mixing in a rising cloudy
parcel. The EMPM can resolve fine-scale variability in the 1D domain down to the smallest
turbulent scales (about 1 mm) and calculate the growth/evaporation of individual cloud droplet
based on each droplet's local environment Su et al. (1998). One unique characteristic of the EMPM
is applying the linear eddy model (Kerstein, 1991) to simulate turbulent deformation and molecular
diffusion separately as an explicit representation of the turbulent mixing process.  Specifically,
turbulent deformation is represented by a sequence of discrete rearrangement events along the 1D
domain, where the scalar field is randomly rearranged using a "triplet map" approach detailed in
(Krueger et al., 1997). Molecular diffusion is calculated with the 1D diffusion equation. With the
capabilities of resolving fine-scale variations and explicitly simulating turbulent mixing, the
EMPM is recognized as a unique and extensively used tool for entrainment and mixing studies (Lu
et al., 2013; Tölle and Krueger, 2014).
To emulate the aircraft measurements using the EMPM, three assumptions are made in this study:
1) entrainment occurs at cloud top; 2) after each entrainment event, the parcel undergoing mixing
descends from cloud top; 3) the virtual aircraft samples sufficient cloudy parcels along a path at
the same height, and those cloudy parcels experience various degrees of entrainment near the cloud
top. The first and second assumptions are satisfied for stratocumulus where the turbulent eddies



and evaporative cooling drives entrainment at cloud top (Wood, 2012). The third assumption is
proposed to mimic the aircraft measurements in real stratocumulus clouds.
The simulation design is illustrated in Fig. 1a. We consider a virtual aircraft that flies at a typical
speed of 100 m s⁻¹ within the cloud, measuring droplet properties at 5 Hz along the leg. Over 2
second interval, the aircraft traverses 200 meters, consisting of 10 in-situ samples, each 20 meters
in length. In the EMPM, each in-situ sample is configured as a one-dimensional domain with a
length of 20 m and the width/depth of 1 mm, resulting in a total volume of 20 cm³ (right panel of
Fig. 1a). The detailed model configuration is shown in Table. 1. The initial droplet number
concentration is set as 80 cm⁻³, consisting of monodisperse haze particles of radius $0.216 \ \mu m$. The
simulation begins with adiabatic lifting of the parcel at a constant velocity of 1 m s⁻¹ until it reaches
the cloud top. The parcel then encounters entrainment, during which subsaturated air replaces a
segment of the cloudy parcel of equal size. The fraction of subsaturated entrained air relative to
the domain size is referred as the entrainment fraction (EF). For instance, Fig. 1a illustrates an
entrainment event with EF of 0.5, meaning that 50% of the cloudy parcel, which is effectively 10
m, is replaced by the entrained subsaturated air. We assume that the entrained dry air is Cloud
Condensation Nuclei (CCN) free thus no CCN is entrained into clouds. After entrainment, the
parcel descends adiabatically at a velocity of -1 m s⁻¹. As the parcel descends, the cloudy air and
the entrained air undergo finite-rate mixing, during which droplets encounter the subsaturated air
and partially or completely evaporate. The number and size of droplets in the domain are updated
at each time step (1s) until all the droplets are completely evaporated.
For each experiment, a total of ten EMPM simulations is conducted with the same initial setting
but with various EFs from 0 to 0.9, representing multiple entrainment events occurring at the cloud
top. Combining all the simulation results produces the collective output illustrated in Fig. 1b. In
this study, we will analyze the output from two perspectives: "bulk" and "local". The bulk-based
perspective emulates the aircraft measurements in clouds, where multiple parcels are sampled at
the same height with each one experiencing distinct entrainment-mixing histories. The local-based
perspective tracks the evolution of cloud microphysical properties in individual parcel after
entrainment, representing the "true" mixing process within the parcel.
To drive the simulations, the idealized thermodynamical profile (Fig. 2) is constructed from the
observations on June 30th, 2017 during the Aerosol and Cloud Experiments in the Eastern North
Atlantic (ACE-ENA) field campaign (Wang et al., 2022). It is noted that a strong inversion layer
exists at 970 m, defining the cloud top height in Table 1. For this study, three experiments (Control,
Dry, and Turbulent) are conducted to represent different entrainment-mixing scenarios. For the
Control simulation, the Eddy Dissipation Rate (EDR) is adapted from the observation as 0.0025
m² s⁻³, representing a typical Sc environment. The thermodynamics of the entrained air is estimated
as the parcel at 10 m above cloud top experiencing adiabatic descent to cloud top. Particularly, the
entrained air temperature and water vapor is estimated as 285.77 K and 8.6 x10⁻³ g/kg. Besides the
control case, two sensitivity cases have been conducted: "Dry" and "Turbulent". For the "Dry"
experiment, the model setup is the same as the control one except the entrained air property is
estimated using the parcel at 20m above cloud top experiencing adiabatic descent to cloud top,



where the air temperature and water vapor is estimated as 288 K and 7.8 x $10^{-3}$ g/kg. The selection
of the entrained parcel distance from cloud top is arbitrary and does not affect the conclusions of
this study. For the "Turbulent" experiment, the model setting is same as the Control one except
the EDR is set as 0.01 $m^2 s^{-3}$, representing a more turbulent environment.

**2.2 Entrainment Mixing Metrics**
With the aircraft measurements, the mixing process is characterized by overlaying the cloud
properties on the mixing diagram and analyzing their collective behaviors (Burnet and Brenguier,
2007; Lehmann et al., 2009; Yum et al., 2015). In this study, the simulation result is displayed in
mixing diagrams similar to those used in the aircraft-measurement studies. In addition, we adapt
the homogeneous mixing degree ($\psi$) to identify the mixing process from the local-based
perspective. The mixing diagram and the associated metrics are introduced in the following.
**2.2.1 $n$-$r^3$ Mixing Diagram**
The $n$-$r^3$ mixing diagram is commonly applied to characterize the mixing process in clouds. In the
diagram, the horizontal and vertical axes represent the normalized number concentration ($n$) and
the average of the third moment of droplet radius ($r^3$). The measurements are normalized by their
theoretical values assuming the cloud parcel ascends adiabatically. For extreme IM, droplet
number is further reduced while the size remains constant, therefore the measurements are
horizontally aligned. For extreme HM, droplet number remains unchanged after dilution, while the
size is reduced due to evaporation. In reality, the mixing can be between the two extreme mixing
types, and thus both droplet number and size may be reduced in the diagram.
**2.2.3 $LWC - \tau_{\text{phase}}$ mixing diagram**
The $L$-$\tau_{\text{phase}}$ mixing diagram was proposed by Yeom et al. (2021) with $x$-coordinates as the
logarithm of liquid water content ($L$) and $y$-coordinates as the logarithm of phase relaxation time
($\tau_{phase}$). $L$ is calculated as:
$$L = \frac{4\pi\rho_L n r^3}{3}$$

where $n$ and $r$ represent the number concentration and droplet radius, and $\rho_L$ is the density of
liquid water.

The phase relaxation time ($\tau_{phase}$) characterizes how rapidly an equilibrium vapor saturation is
reached by evaporation of a population of droplets (Lehmann et al., 2009; Jeffery and Reisner,
2006). For the EMPM simulation output, $\tau_{phase}$ is calculated following the method applied in
Tölle and Krueger (2014):
$$\tau_{phase} = \frac{1}{4\pi D_v N} \frac{R_v + a}{R_v^2}$$





where $N$ and $R_v$ represent the domain-mean droplet number and radius estimated at the time
immediately following the entrainment event. $D_v$ is the molecular diffusivity of water vapor and
is taken as 0.256 cm$^2$s$^{-1}$. $a$ is the accommodation length taken as 2 $\mu m$.

To interpret the $L$- $\tau_{\text{phase}}$ mixing diagram, linear regression is performed between the logarithm
of $L$ and $\tau_{\text{phase}}$ dataset and the corresponding slope is used for mixing classification: the slope of
–1 represents extreme IM, while the HM should asymptote to the line with slope of –1/3.

**2.2.3** Homogeneous mixing degree
Based on the $n$-$r^3$ mixing diagram, Lu et al. (2013) proposed the homogeneous mixing degree
following the calculation:
$$\beta = \tan^{-1}\left(\frac{\frac{r_v^3}{r_{va}^3} - 1}{\frac{n}{n_a} - \frac{n_h}{n_a}}\right)$$

where $r_v$ and $r_{av}$ represent the volume-mean radius and the adiabatic radius of droplets, $n$ is the
number concentration, $n_a$ is the adiabatic number concentration, $n_h$ is the number concentration
immediately following the entrainment event but prior to evaporation and accounts for the dilution
by entrainment; The parameter $\beta$ effectively calculates the angle, with unit of radian, from the
extremely IM line (detailed illustration is shown in Fig. 1 in Lu et al. (2013)).
$\beta$ is commonly normalized by $\pi/2$ to represent the homogeneous mixing degree ($\psi$):
$$\psi = \frac{\beta}{\pi/2}$$

$\psi$ ranges from 0 to 1, with large values indicating IM and small $\psi$ indicating HM. Since $\psi$ is
estimated upon each parcel instead of a collective datapoints, we apply $\psi$ to characterize the local
mixing process within the parcel.



**3.  Results**
**3.1 Cloud Properties from the EMPM simulation**
The simulated domain-averaged cloud properties under various entrainment events are shown in
Fig. 3. When the parcel ascends adiabatically, the LWC linearly increases from cloud base (i.e.
745 m) to cloud top with the maximum value of 0.42 g m$^{-3}$ (red line in Fig. 3a). The domain-
averaged cloud droplet radius increased to 10.7 um (red line in Fig. 3b). Correspondingly, a total





of 1600 droplets is activated at cloud base and the number remain unchanged towards cloud top.
Considering the EMPM domain of 20 cm$^3$, the number concentration within the undiluted
ascending parcel is 80 cm$^{-3}$. As introduced in Sec 2.1, the parcel descent immediately after reaching
cloud top. When no entrainment occurs at the cloud top, the simulated cloud properties within the
descending parcel is shown as the blue line in Fig. 3. It is noticed that LWC and the droplet radius
do not follow the trajectory of the ascending parcel but with slightly enhanced value. This
enhanced radius/LWC is caused by the hysteresis effect manifested as the time-lag adjustment of
the parcel supersaturation responding to the change of dynamics (Yang et al., 2018). Specifically,
as the parcel starts moving downward as a consequence of the changed velocity from 1 ms$^{-1}$ to -1
ms$^{-1}$, the supersaturation within the parcel remains positive with value of 0.47 % (red line in Fig.
3d). Consequently, the droplet continues to grow until the supersaturation is removed. It is shown
that the supersaturation turns to negative at the height of 943m, which is 7 m down from cloud top.
The extra growth over this 7m distance led to a larger LWC and radius in the downward branch
(Fig. 3a, c).
For the descending parcels with various entrainment events, LWC and droplet number reduce
instantaneously at cloud top (Fig. 3a, c) due to the replacement by entrained air. Meanwhile, the
domain-mean radius remains constant at cloud top (Fig.3 b) as the evaporation-mixing process has
not yet begun. As the parcel descends, LWC, droplet radius and number decrease due to
evaporation. The extent of the reduction depends on the entrainment fraction. For strong
entrainment event, the mixed parcel is much drier thus experiencing stronger evaporation, leading
to lower LWC, smaller radius, and fewer droplets. Under large EF, droplets within the parcel are
completely evaporated at a higher altitude. For instance, for the EF of 0.4 (black line in Fig. 3),
droplets are evaporated at 862 m, which is 88 m below the cloud top (Fig. 3c).

## 3.2 Entrainment Mixing Behavior within Clouds
### 3.2.1   Bulk Perspective

We use the two mixing diagrams to analyze the EMPM simulations from the aircraft-based
perspective. In the $n$-$r^3$ mixing-diagram (Fig. 4a, c, e), droplet number and $r^3$ are normalized by
the value in the descending parcel without entrainment occurring (blue line in Fig. 3b, c). For the
control experiment (Fig. 4a), the collective behavior of the 10 simulations with different EFs shows
reduced droplet number but unchanged radius at 5m below cloud top (circles in Fig. 4a). The
reduced number is caused by the entrainment when a given fraction of the domain is
instantaneously replaced by the droplet-free air. At 5m below cloud top, droplets have not yet
experienced strong evaporation because only 5 s has elapsed since the entrainment event. To better
visualize the mixing signature at different heights, polynomial lines are fitted based on the
normalized $n$-$r^3$ diagram. The fitted line at 5 m below cloud top is horizontally aligned reasonably
well with the normalized $r^3$ = 1 (black line in Fig. 4a), exhibiting a typical IM signature. This IM
phenomenon is echoed in the $L$-$\tau_{phase}$ mixing diagram: the slope of the linear regression of the
datasets at 5 m below cloud top is -0.81 (circles in Fig. 4b)., which is close to the IM reference line
with the slop of -1.





As the parcels descend deeper into the cloud, those with different EFs exhibit distinct evaporation
histories, leading to contrasting mixing signatures. Taking the control experiment at 50 m below
cloud top (squares in Fig. 4a) as an example, the normalized $r^3$ is reduced to 0.48 for the parcel
with EF equals 0.8, while the normalized $r^3$ is 0.92 for the parcel with EF equals 0.1. As a result,
the collected behavior of all the parcels at this level exhibit HM signatures (red line in Fig. 4a)
with reduced droplet numbers and radii. It is further noted that the HM signature is more prominent
deeper into the cloud (i.e., further away from the cloud top). Comparing the fitted lines from two
height levels (red and blue lines in Fig. 4a), parcels at 200 m below cloud top show greater
reduction of radius compared to the parcels at 50 m below cloud top. This transition of the mixing
signatures is more evident in the $L$-$\tau_{phase}$ mixing diagram (Fig. 4b). As the distance from the cloud
top increases, the collective datapoints rotate counterclockwise from the IM (red line) to the HM
(blue line) reference line. Specifically, for heights at 5m (circles), 50m (squares) and 200m
(triangles) from cloud top, the slopes of the linear regression are -0.81, -0.32 and -0.28, exhibiting
a stronger HM degree deeper into cloud.
The two sensitivity experiments (i.e., Dry and Turbulent) lead to similar conclusions as the Control
one with slightly different behavior. When the entrained air is drier, the mixed parcel experiences
stronger evaporation thus exhibiting a small degree of HM signature near cloud top. In the Dry
experiment (Fig. 4b), the normalized $r^3$ at 5 m below cloud top decreases by 17% in the simulation
with an EF of 0.9, causing the fitted line to bend downward toward smaller radii in the large EF
regime (black line in Fig. 4b). However, it is still clear that near cloud top the mixing is
predominantly IM with a significant reduction of droplet number and a small reduction of radius.
This IM-dominated signature is also identified in the $L$-$\tau_{phase}$ mixing diagram (Fig. 4d) in which
parcels near cloud top (circles) align well with the IM reference line (red dashed line). In the
Turbulent experiment (Fig. 4e, f), the mixing signature is similar to the Control one near cloud top
but shows differences deeper into the cloud. For a given normalized $n$, the Turbulent experiment
is characterized by a greater reduction of radius compared to the Control one. For instance, at 200
m from cloud top, where the normalized $n$ equals 0.6, the normalized $r^3$ for the Control and
Turbulent experiments are 0.7 and 0.48, respectively. This large reduction of droplet size is
expected as strong turbulence favors efficient mixing and enhance the HM signature.
Overall, despite the different thermodynamics and dynamics of the entrained air, simulations show
a clear IM feature near cloud top and HM within the cloud, with greater degree of HM deep into
the cloud. These model-based results are consistent with the aircraft measurements in Sc (Yum et
al., 2015; Yeom et al., 2021), thus providing a strong foundation for more detailed investigations
in the next section.





### 3.2.2 Local Perspective


In this section the EMPM simulations in Sec 3.2.1 are interpreted from the local-based perspective
as introduced in Fig. 1. Specifically, instead of analyzing parcels with different EFs at given height,
we evaluate the mixing process of each parcel by tracking its history. Figure 5a shows the $n$-$r^3$
mixing diagram for four parcels with EF of 0.1, 0.3, 0.5, and 0.7. The parcels initially follow a
near-vertical path (i.e., indicating a reduction in droplet size with minimal change in number
concentration) near the cloud top, then gradually tilt toward the smaller number regime. These
features show HM near cloud top and the mixing more tends to inhomogeneous deeper into cloud.
The strongest HM signature is observed for the parcel with EF = 0.1 (blue symbols), where at 50
m below the cloud top, the normalized $r^3$ decreases by 18%, while the normalized number
decreases by only 1.5%. This behavior highlights the dominance of HM near cloud top.
To quantitatively describe the mixing process in each parcel, we adapt the homogeneous mixing
degree $\psi$ proposed by Lu et al. (2013). As introduced in Sec 2.2, $\psi$ is evaluated based on the $n$-$r^3$
mixing diagram by calculating the relative changes of droplet size and number after each mixing
event. Since estimating $\psi$ only requires the change of cloud microphysics within each parcel, it is
suitable to illustrate the mixing process from the local perspective. For the four selected parcels,
$\psi$ consistently decreases from cloud top to base (Fig. 5b). As $\psi$ =1 indicates extremely HM, the
large $\psi$ at Fig. 5b indicates strong HM at cloud top.  Deeper into the cloud, $\psi$ decreases, indicating
a weakening of HM and an increasing influence of IM. This behavior holds true for all the four
simulations regardless of the entrainment degree. Parcel with EF 0.1 has the largest $\psi$ throughout
the cloud and exhibits the most pronounced HM signature. Parcel with EF of 0.3 and 0.5 have $\psi$
decreasing from 1 to 0.65 and 0.76 at 100 below cloud base.
The HM–IM transition observed from the local perspective appears to contradict the mixing
behavior suggested by the bulk perspective. We propose that this inconsistency arises from the
differing analytical perspectives. The local perspective indicated in Fig. 5 follows the continuous
evolution of individual parcel, revealing the "true" mixing processes. While the bulk perspective
captures a "snapshot" of an ensemble of parcels, each with distinct entrainment and mixing
histories. Near the cloud top, the entrained air replaces the cloudy air and instantaneously reduce
the droplet number. Immediately following entrainment, parcels with large EF experience larger
reductions of droplet number, while evaporation has not yet efficient enough to reduce droplet size.
Thus, a collection of multiple parcels with different entrainment events generates an IM signature.
As the parcel descends deeper into the cloud, mixing with dry air continues and evaporation
becomes efficient, leading to a reduction in droplet size. Parcels with larger EF experiencing
stronger evaporation and results in more pronounced decrease in droplet size and number.
Consequently, a collection of parcels with different EF tends to exhibit a HM signature deeper into
the cloud.
Based on this reasoning, we further propose that from the bulk perspective, mixing is always
manifested as IM near cloud top and HM towards cloud base, regardless of the mixing process
exhibited from local perspective. To testify this hypothesis, we conduct a strict IM experiment
with the same configuration as the Control experiment but setting an extremely low EDR value of
$10^{-14}$ $m^2$ $s^{-3}$. This nonrealistic EDR value results in low mixing efficiency in the EMPM simulation



where 100 steps of diffusion (e.g. evaporation) are performed per turbulent mixing step. As a comparison, for the "Turbulent" experiment where EDR is 0.01 m$^2$ s$^{-3}$, the EMPM performs 100 mixing steps per diffusion step. Thus, the conducted IM experiment ensures strict IM scenario with evaporation much faster than the turbulent mixing.

The mixing process of the strict IM experiment from the local perspective is shown in Fig. 6. In the $n$-$r^3$ mixing diagram, the parcel experiencing greater reduction of number compared with radius. Take the simulation with EF of 0.1 (blue symbol in Fig. 6a) for example, from 2m to 150m from cloud top, droplet number is reduced by 6% while the normalized $r^3$ is only reduced by 0.8%. The evolution of $\psi$ within clouds (Fig. 6b) indicate an IM-HM transition from cloud top to base. Specifically, $\psi$ increase from 0 to approximately 0.4 through the clouds, suggesting strong IM feature near cloud top and an increase degree of HM at lower levels. The negative $\beta$ near cloud top is caused by the growth of droplet after entrainment, which may be caused by the remaining supersaturated environment at cloud top as discussed in Fig. 3d.

Although strong IM signature is identified for each parcel, the collective behavior of multiple parcels still exhibits IM near cloud top and HM within cloud. At 2m below cloud top, parcels with various EFs are aligned horizontally (circles in Fig. 6a) and is manifested as IM signature. At 150 m below the cloud top, stronger entrainment events lead to greater reductions in droplet radius. For the parcel with EF = 0.7 (yellow symbols), the normalized $r^3$ decreases by 13%, whereas for the parcel with EF = 0.1, the reduction is only 0.8%. As a result, connecting the parcels at 150m below cloud top (inverted triangles in Fig. 6a) reveals HM signature. It is noticeable that the reduction of droplet size in Fig. 6a is significantly smaller than the control experiment as shown in Fig. 5a. This difference is expected as the turbulent mixing is strongly inhibited in Fig. 6a, thus the entrained dry air cannot efficiently mix with cloudy air, which eventually inhibits evaporation of droplets. Nevertheless, the experiments shown in Fig. 5 and Fig. 6 demonstrate that, from the bulk perspective, mixing behavior consistently exhibits IM near the cloud top, with an increasing signature of HM deeper within the cloud, regardless of the local mixing processes occurring in individual parcels.

### 3.3 Isobaric-Mixing Experiment

In previous sections, we have reconstructed the mixing behavior in Sc using EMPM simulations which is consistent with the aircraft-based measurements. However, the non-isobaric mixing process in previous experiments may lead to ambiguity for mixing interpretation. Specifically, when droplets evaporate in a descending parcel, the subsaturated environment can be caused by adiabatic warming and non-isobaric mixing. To isolate these two effects, we conduct an isobaric mixing experiment. The experiment setup is the same as the control one except after entrainment event near cloud top, the parcel velocity is set to 0 m s$^{-1}$. This setting ensures the parcel only experiencing isobaric mixing after the entrainment at cloud top.



Fig. 7 shows the mixing diagrams at three elapsed times after the entrainment event. At 3s, parcels
with different EFs are closely aligned with the line of normalized $r^3 = 1$. Correspondingly, the
slope of the fitted line in the $L$-$\tau_{\mathrm{phase}}$ diagram is -0.81 (Circle in Fig. 7b). These two features
suggest IM at the beginning of mixing process. At 15s, HM signature is identified with parcels of
large EF experiencing greater reduction of radii and number (red line in Fig. 7a). At 90s, stronger
reduction of droplets size and number indicating a more prominent HM signature (blue line in Fig.
7a). The $L$-$\tau_{\mathrm{phase}}$ diagram echoes the stronger HM feature as mixing continuing with the fitted
slope increases from -0.56 to -0.42 from 15s to 90s.
To better illustrate the mixing process as a function of time, the normalized standard deviation of
water vapor is plotted for four experiments (Fig. 8a). Specifically, the standard deviation of water
vapor ($\delta_{qv}$) is calculated at each time step within the 1D domain with a domain size of 20 m and
grid size of 1 mm. Then normalization is performed by dividing $\delta_{qv}(t)$ by $\delta_{qv}$ at 1 s after
entrainment. The evolution of the variances of $\delta_{qv}$ can illustrate the mixing time scale (Tölle and
Krueger, 2014). In Fig. 8a, $\delta_{qv}$ is maximum after the entrainment. As time goes by, $q_v$ decreases
as mixing occurs between the entrained air and cloudy air. Parcels with the small EF experience a
short mixing time compared with those with large EF. For instance, the parcel with EF 0.3 needs
60 s to reach the equilibrium state (green line in Fig. 8a) while the one with EF 0.1 needs only 20
s (blue line in Fig. 8a) to homogenize water vaper within the domain.
The parcel-based mixing behavior for four parcels is shown in Fig. 8b. Tracking individual parcels,
it is clearly shown that the parcel experiencing HM has a greater reduction of radii compared to
number. The most extreme case is for the parcel with EF of 0.1 (blue symbols in Fig. 8b): during
the mixing process the normalized $r^3$ decreases by 17% while the normalized number barely
changes. Correspondingly, the $\psi$ parameter decreases from 1 to 0.97 from 0 to 12 s, indicating
extreme HM. Parcels with large EF exhibit HM signatures after a greater duration of mixing. For
instance, $\psi$ for the parcels with EF of 0.3 and 0.4 decrease from 1 to 0.7 at 80 s after entrainment,
suggesting the HM signature is enhanced as the mixing proceeds.
In a nutshell, the isobaric mixing experiment exhibits similar results as shown in the previous
experiments. The collective mixing behavior of multiple parcels exhibits IM at the beginning of
mixing and HM at later time. The elapsed time in the isobaric mixing experiment is equivalent to
the distance from cloud top for the non-isobaric mixing experiments. The true mixing process as
indicated from the local-based perspective, on the other hand, may be completely different from
the collective mixing behavior. This isobaric mixing experiment reinforces the conclusion that the
IM-HM transition from the bulk perspective results from the sampling strategy in clouds rather
than true mixing process in the parcel.








## 4. Discussion

Using multiple EMPM simulations, we successfully reproduce the commonly aircraft-observed result with IM near cloud top and HM within cloud. We further explain this phenomenon in Fig. 9. The aircraft measurements include multiple cloud parcels experiencing different entrainment-mixing histories. In Sc, entrainment occurs at the cloud top where a horizontal fraction of cloudy air is replaced by the free atmosphere. If the entrained air is cloud droplet-free, the entrainment event instantaneously reduces droplet number. Parcels experiencing strong entrainment have greater reductions of droplet number. The moment after entrainment, the dry air has not yet mixed with cloudy air, which is required to generate strong evaporation, thus the domain-averaged size remains constant. A collection of multiple parcels at cloud top are aligned along a horizontal line indicating the IM signature. As the parcel descends into cloud, mixing and evaporation occur collectively to reduce droplet size and number. Parcels with strong entrainment at the cloud top are associated with large entrainment fraction, resulting in a drier environment compared to parcels with smaller EF. Deeper into the cloud, parcels with large EF experience stronger evaporation, leading to a greater reduction in both droplet size and number. The collective view of parcels with different EFs in the $n$-$r^3$ mixing diagram exhibits HM signature.

This explanation is essentially consistent with the "vertical circulation" hypothesis proposed by Yum et al. (2015). In this study we use the EMPM simulations for a thoughtful demonstration and aim to raise the awareness of this modeling approach for investigating entrainment-mixing processes. Particularly, the aircraft measurements should be interpreted with caution especially when multiple samples along the aircraft traverse are overlapped in the mixing diagram. The collective behavior of different samples at given altitude may exhibit a result which does not represent the true mixing mechanism of each sample. For the similar reason, the Large Eddy Simulation (LES) output should also be analyzed with caution. Collecting cloud properties along multiple grids at a given height in the model generates pseudo "aircraft-based measurements", which may also lead to misinterpretation of the mixing process. Lagrangian-based models, with the capability of tracking the history of each parcel, should serve as a more suitable tool for mixing investigations (Hoffmann and Feingold, 2019; Lim and Hoffmann, 2024). From the observational perspective, while the Lagrangian-based tracking approach is not applicable, alternative measurement methods developed in recent decades is helpful to mitigate the mixing artifacts generated from the aircraft measurements. For instance, the Cloudkite platform deployed at the kite-stabilized balloons (Schröder, 2023) and the holographic imaging technique (Beals et al., 2015) can provide high spatio-temporally resolved measurements down to cm-scales. Such fine-resolution observations capture the local cloud mixing state more representatively, offering deeper insights into the entrainment mixing processes within clouds.



## 5. Conclusion

In this study, we conduct EMPM simulations to understand the entrainment-mixing process observed from aircraft measurements in stratocumulus cloud. Three experiments are conducted with different thermodynamic and turbulence environments. Each experiment consists of ten simulations, with each simulation representing a 20-meter parcel undergoing various entrainment degree at cloud top and distinct mixing history. The overall entrainment-mixing process for the simulations is analyzed from two views: the bulk-based and local perspective. The bulk perspective resembles the aircraft measurements in clouds and is illustrated by two commonly used mixing diagrams. The local perspective reflects the true mixing behavior in each parcel and is quantified by the homogeneous mixing degree ($\psi$) developed by Lu et al. (2013).

From the bulk perspective, the simulated mixing is identified as IM near cloud top and HM within cloud, which is consistent with the aircraft measurements in real clouds. However, this vertical progression primarily arises from the collective view of multiple parcels experiencing different mixing stages, in which strong evaporation in some parcels juxtapose with weak evaporation in others. This bulk view obscures the parcel's actual mixing process and leads to the appearance of a systematic IM-HM transition within cloud, even in cases where the underlying local mixing within each parcel could be substantially different. It is suggested that future mixing investigations in clouds should carefully re-examine the aircraft-based interpretation and consider incorporating Lagrangian approaches.

It is noted that the purpose of this study is to urge caution when interpreting aircraft measurements and LES simulations in entrainment–mixing research. This study does not aim to conclude the entrainment mixing behaviors in clouds. To advance the understanding of mixing processes in real clouds, emerging measurement technologies, such as the holographic detectors and tethered platforms, offer critical insights to observe mixing at the parcel scale. Additionally, for illustrative purposes, this study employs an idealized mixing framework in which each parcel evolves independently, with no mixing between parcels with differing entrainment histories. While a more sophisticated mixing scheme could better approximate observational realities, such complexity falls outside the scope of the present work.






**Code/Data availability:**

The EMPM codes used in this study is available upon request from the authors.

**Financial Support:**

Z. Zhu, F. Yang, Y. Liu were funded by the Department of Energy (DOE) as part of the
Atmospheric System Research (ASR) program under Contract DE-SC0012704. S. Krueger was
supported by NSF grant AGS-2133229.

**Author contribution:**

ZZ designed the methodology and carried out the analysis. FY contributed to the study design. SK
provided guidance on the use of the EMPM. YL assisted with the interpretation of results. ZZ
drafted the manuscript, with all co-authors contributing to revisions and editing.


**Competing interests:**

The corresponding author has declared that neither they nor their co-authors have any competing
interests.














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



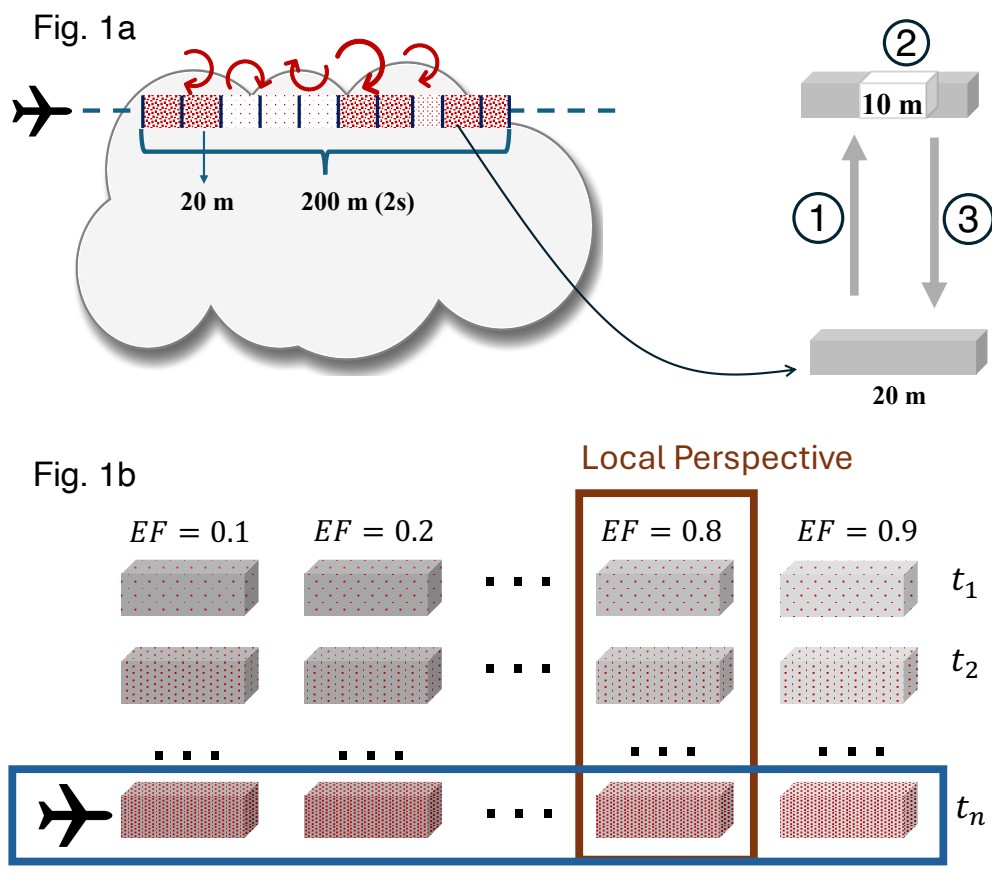


Figure 1: Illustration of experiment set up: a) the left panel illustrates aircraft measurements near cloud top. During 2s, the aircraft traverse 200 m, acquiring 10 samples. Each sample corresponds to a 20-m cloud parcel, which is simulated by the EMPM. The sampled cloud parcels exhibit varying entrainment fractions as indicated by the shading. Lighter (sparser) shading corresponds to samples with higher entrainment fractions. The right panel illustrates the simulated parcel experiencing three stages: ① rising, ② entrainment and ③ sinking. b) Illustration of the local and bulk-based perspective for simulations analysis: the local perspective tracks the change of properties with time after the entrainment events; the bulk perspective collects multiple parcels with various EF at a given time. $t_1$ represents the entrainment moment, $t_n$ represents an arbitrary time step after entrainment.










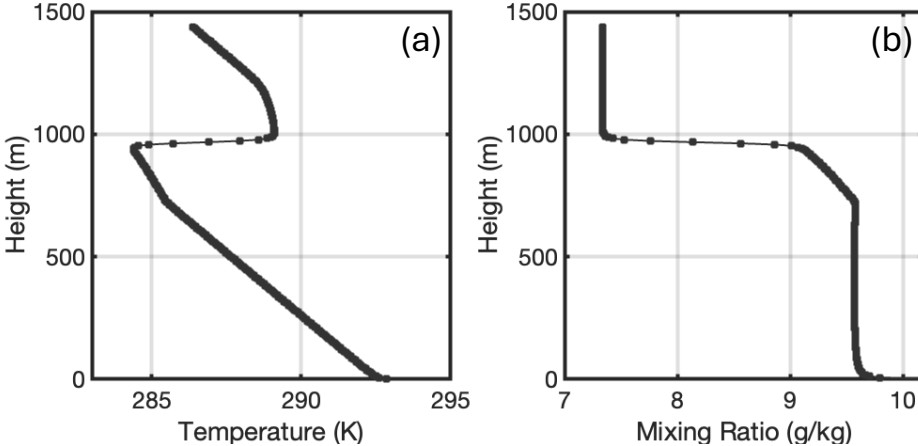


Figure 2: The idealized a) temperature and b) mixing ratio profiles based on the sounding observation at 5:30 UTC on June 30th, 2017 during the ACE-ENA field campaign.







Table1 – Model Configuration

| Parameter | Control | Dry | Turbulent |
|---|---|---|---|
| Domain Length (m) | | 20m | |
| CCN Concentration (cm$^{-3}$) | | 80 | |
| Vertical Air Velocity (ms$^{-1}$) | | $\pm1$ | |
| Cloud Top Height (m) | | 950 | |
| Aerosol Size Distribution | | Monodisperse | |
| Initial mass of droplet solute (kg) | | $0.1122*10^{-17}$ | |
| Initial aerosol radius (m) | | $0.216*10^{-6}$ | |
| Type of aerosol | | NaCl | |
| Eddy Dissipation Rate (m$^2$s$^{-3}$) | 0.0025 | | 0.01 |
| Entrained air temperature (K) | 285.77 | 288 | 285.77 |
| Entrained air water vapor(g/kg) | $8.6*10^{-3}$ | $7.9*10^{-3}$ | $8.6*10^{-3}$ |

Table 1: Model configurations for the control, dry and turbulent simulation experiment.






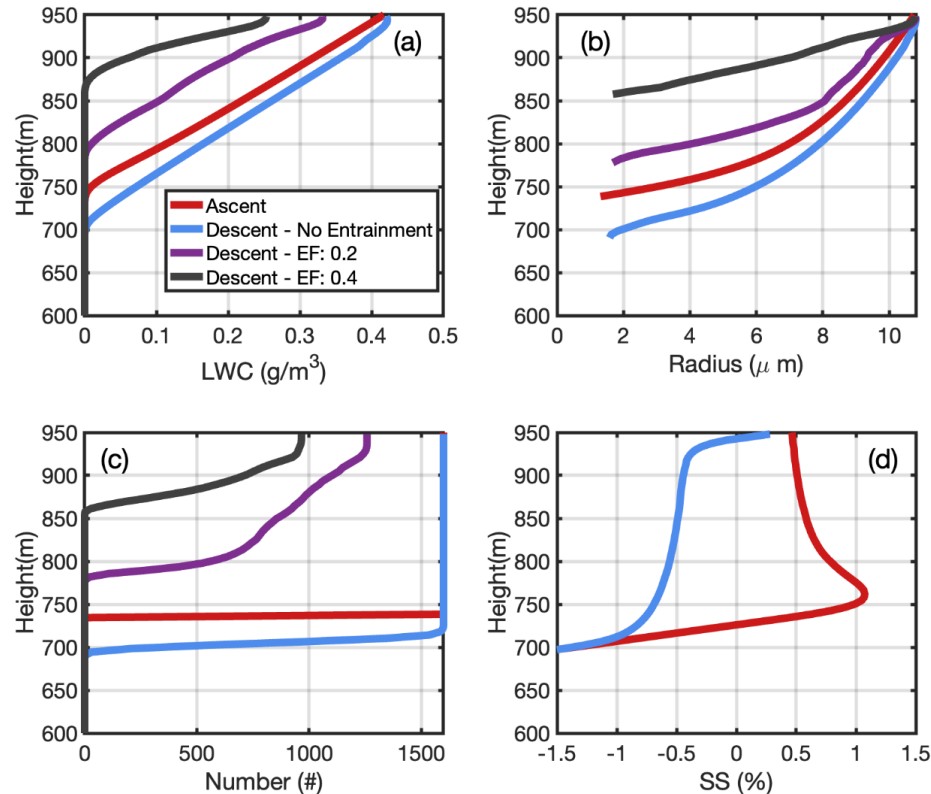


Figure 3: For the control experiment, domain-averaged cloud properties as function of height: a)
LWC, b) radius, c) droplet number d) supersaturation. The red line represents the ascending parcel,
while the blue, purple and black lines represent the descending parcel with entrainment fraction of
0, 0. 2 and 0.4 respectively. In (d) only the ascending parcel and the descending parcel with EF of
0. 4 is shown.










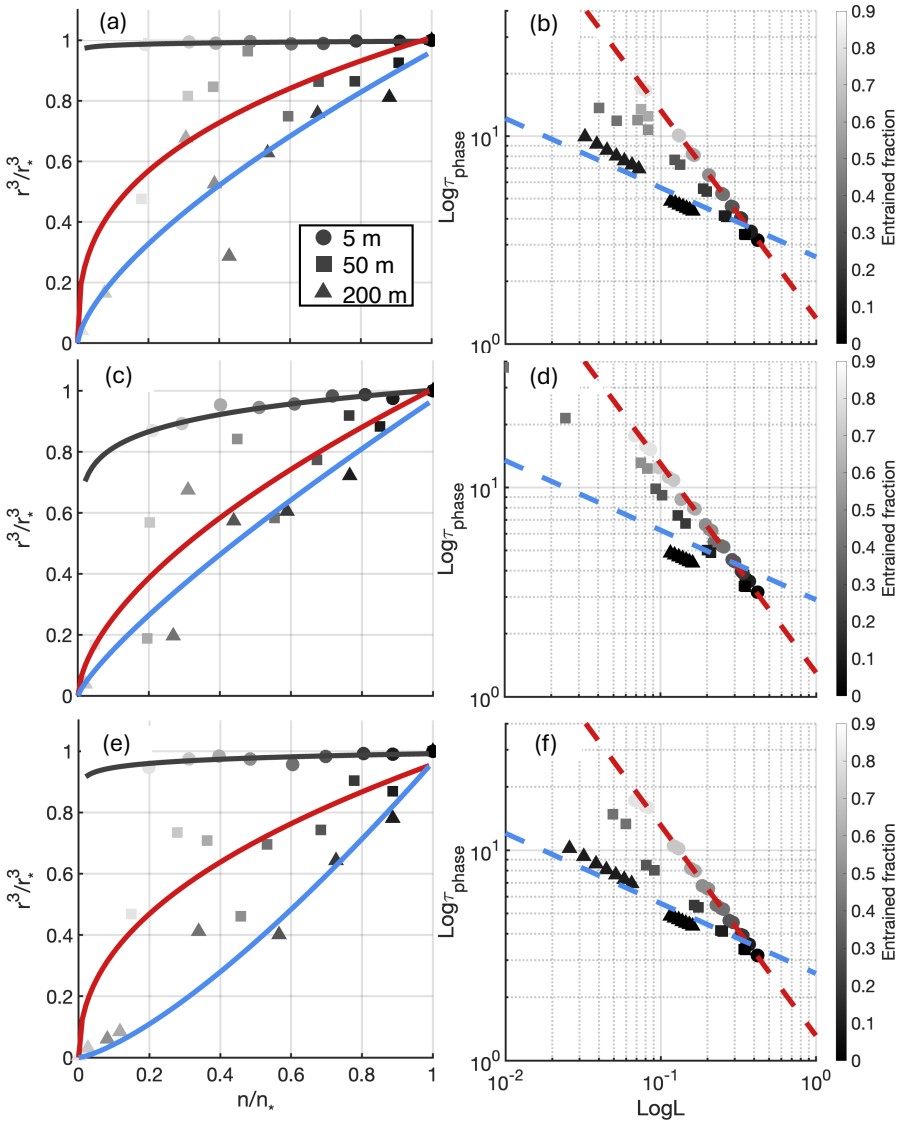


Figure 4: Mixing diagrams for three experiments: The left panel represents the $n$-$r^3$ mixing
diagram for the (a) control, (c) dry and (e) turbulent experiment. The circle, square and triangle
represents simulations at 5, 50 and 200m from cloud top. The black, red and blue lines represent
the polynomial fitting of the parcels at each height level. The right panel indicates the $L$-$\tau_{phase}$
mixing diagram for the (b) control, (d) dry and (f) turbulent experiment. The circle, square and
triangle represents simulations at three heights as indicated in (a). The red, blue dashed line
represents the IM and HM reference line with slope of -1 and -1/3.





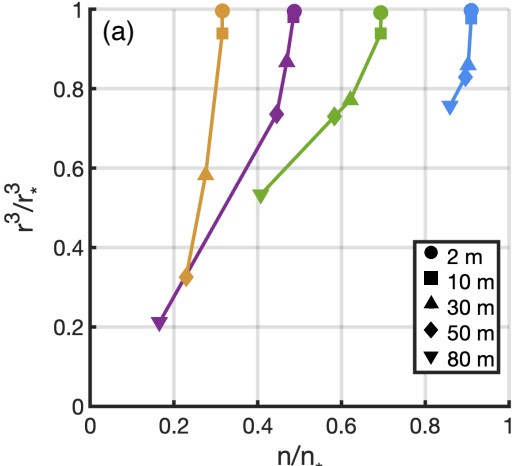
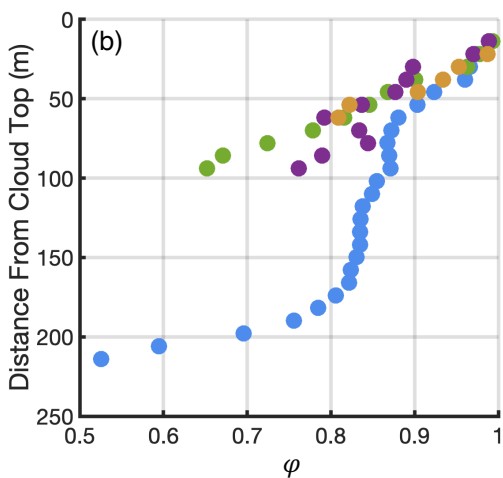

646

Figure 5: a) $n$-$r^3$ mixing diagram from the parcel-based perspective for the control experiment.
The circle, square, triangle, diamond and the reverse-triangle indicate the height of 2m, 10m, 30m,
50m and 80m from cloud top. The blue, green, purple and yellow represents the parcel with EF of
0.1, 0.3, 0.5 and 0.7. b) The homogeneous mixing degree ($\varphi$) as a function of distance from cloud
top, different color represents parcel with different EF as indicated in (a).

















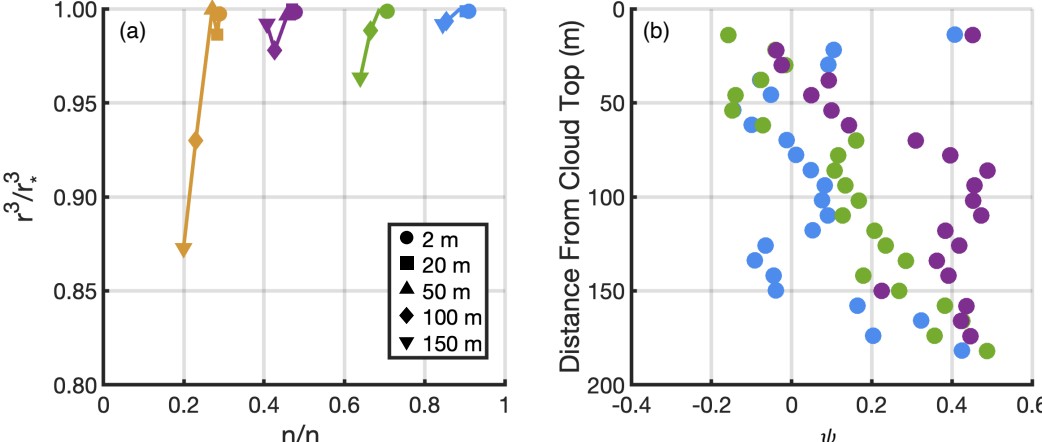


Figure 6: a) $n$-$r^3$ mixing diagram from the parcel-based perspective for the strict IM experiment.
The circle, square, triangle, diamond and the reverse-triangle indicate the parcel at height of 2m,
20m, 50m, 100m and 150m from cloud top. The blue, green, purple and yellow represents parcel
with EF of 0.1, 0.3, 0.5 and 0.7. b) The homogeneous mixing degree ($\varphi$) as a function height for
the strict IM experiment. The blue, green, purple color represents parcel with EF of 0.1, 0.3, 0.5.










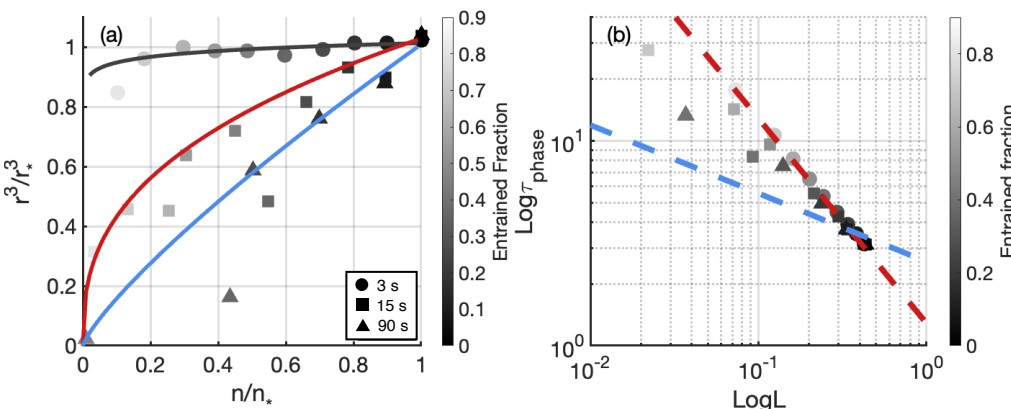

Figure 7: (a) $n$-$r^3$ mixing diagram from the bulk perspective for the isobaric mixing experiment:
The circle, square and triangle represents the elapsed time of 3s, 15s and 90s after entrainment.
The black, red and blue lines represent the polynomial fitting for the parcels at 3s, 15s and 90s,
respectively. (b) $L$-$\tau_{\text{phase}}$ mixing diagram for the isobaric mixing experiment. The circle, square
and triangle represents the elapsed time at 3s, 15s and 90s after entrainment. The red, blue dashed
line represents the IM and HM reference line with slope of -1 and -1/3.






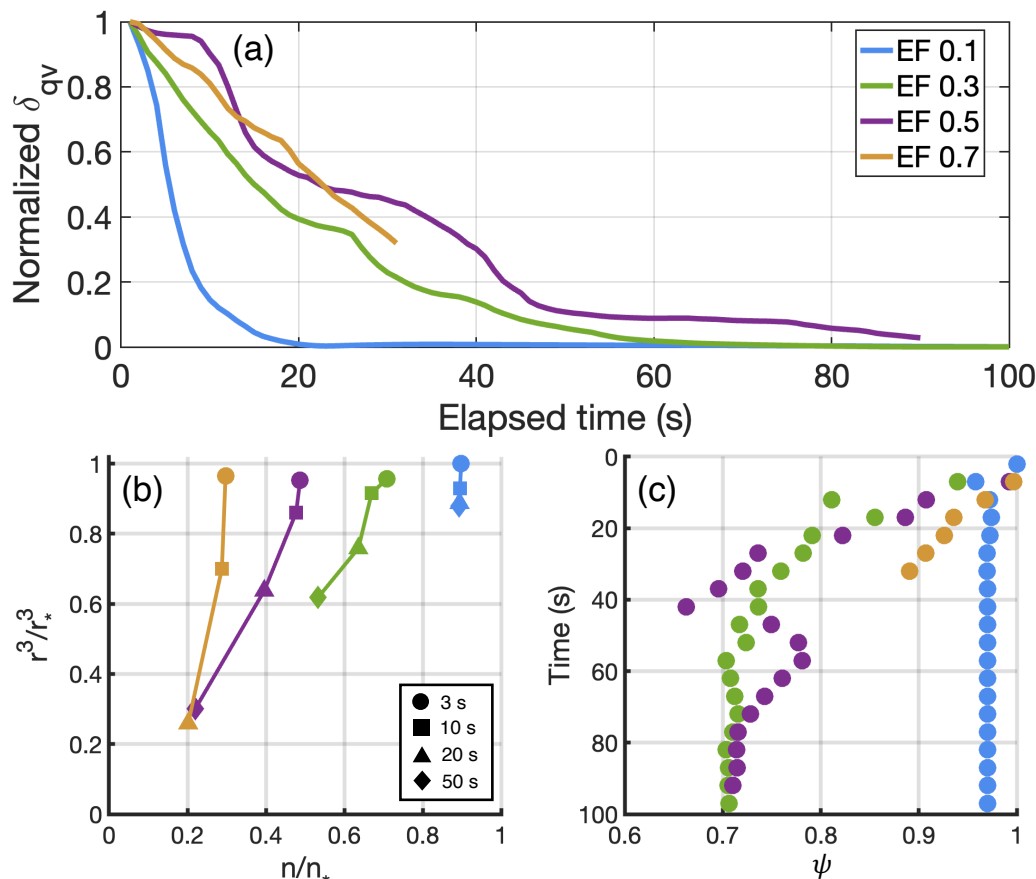


Figure 8: (a) Normalized standard deviation of water vapor ($\delta_{qv}$) in the parcel after entrainment
for the isobaric mixing experiment. The blue, green, purple and yellow line represents the parcel
with EF of 0.1, 0.3, 0.5 and 0.7. (b) $n$-$r^3$ mixing diagram for the isobaric mixing experiment. The
blue, green, purple and yellow symbol represents parcel with EF of 0.1, 0.3, 0.5 and 0.7. The circle,
square, triangle and diamond indicate the parcel at elapsed time of 3s, 10s, 20s, 50s after
entrainment. (c)The homogeneous mixing degree ($\varphi$) as a function elapsed time for the isobaric
mixing experiment. Different color represents parcel with different EF indicated in (b).






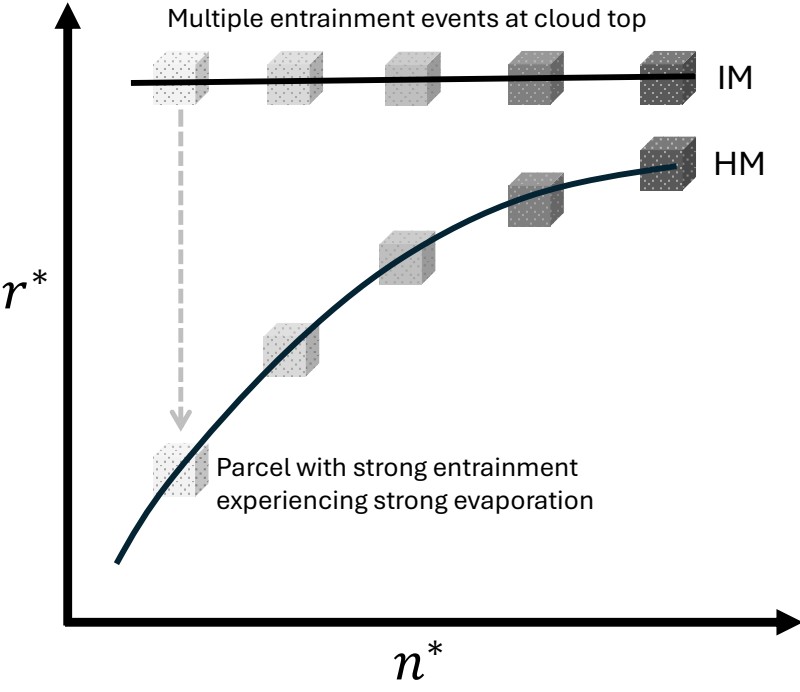


Figure 9: Illustration of the IM-HM transition within Sc from the bulk perspective. Parcels with darker (lighter) shading corresponds to samples with lower (higher) entrainment fractions. The horizontal black line represents the IM behavior occurring near cloud top, the curved line represents the HM behavior occurring within cloud.









