# Peer review of "Why Is Height-Dependent Mixing Observed in Stratocumulus?"

_EGUsphere, 2025_

## Referee Comment (RC1)

**Review of *"Why Is Height-Dependent Mixing Observed in Stratocumulus?"***

This manuscript investigates entrainment–mixing processes in stratocumulus using the Explicit Mixing Parcel Model (EMPM). By emulating virtual aircraft measurements, the authors argue that the frequently observed transition from inhomogeneous mixing (IM) near cloud top to homogeneous mixing (HM) within cloud depth is essentially a collective behavior of multiple parcels sampled at the same height, experiencing distinct entrainment–mixing–evaporation histories, rather than reflecting the true local mixing mechanism. The study compares bulk versus local perspectives, introduces isobaric mixing simulations, and provides a discussion on the implications for interpreting both aircraft and LES data. The topic is highly relevant to the cloud physics community, and the work provides some insight into entrainment–mixing interpretation. However, the revisions are needed to clarify assumptions, better explain contradictory perspectives, and explicitly define terms such as "near cloud top." With these improvements, the paper will be a valuable contribution to the literature.

**Major comments:**

1. CCN concentration in entrained air: The manuscript assumes that the entrained dry air is CCN-free. This is a strong simplification, since in reality entrained air frequently contains at least some aerosols that can serve as CCN. I strongly encourage the authors to either (a) perform additional sensitivity experiments with non-zero CCN concentration, or (b) explicitly discuss how this assumption may bias their results and conclusions. Without such treatment, the applicability of the EMPM findings to real stratocumulus environments remains limited.

2. Descending velocity after entrainment: A uniform descent rate of -1 m s$^{-1}$ is imposed. However, the actual descent speed is likely to vary with entrainment fraction (EF), local turbulence intensity, and thermodynamic structure. The authors should justify this choice more thoroughly and, ideally, include sensitivity tests with varying descent rates. A clear discussion of this limitation is necessary to describe how robust the reported IM–HM transition is under different dynamical conditions.

3. The local perspective suggests HM near cloud top transitioning to IM deeper in cloud, whereas the bulk view shows the opposite. This apparent contradiction is central to the study but is not explained clearly enough. A related issue is that in several analyses (e.g., Figs. 4-6), the term "near cloud top" is used without a quantitative definition. Since the results depend sensitively on how close the sampling is to the inversion, the lack of a clear threshold (e.g., within 5 m or 10 m below cloud top) makes the interpretation appear ambiguous.

**Minor comments:**

1. Ensure that Sc is always defined as stratocumulus upon first use and then used consistently.

2. Lines 17 and 78: "IH" seems to be a typographical error and should be corrected to 'IM'.

3. Line 187: What is the accommodation length 2μm? Please explain it.

4. Figures 4–6: Clear annotations (e.g., "IM-like" vs. "HM-like") on the fitted lines might be helpful to understand these diagrams.

---

## Referee Comment (RC2)

**Reviewer comments on *Why Is Height-Dependent Mixing Observed in Stratocumulus?**

This manuscript examines how to interpret homogeneous versus inhomogeneous mixing (HM/IM) signatures in stratocumulus, integrating insights from in-situ observations and modeling with a framework that explicitly resolves inhomogeneous mixing. This manuscript provides a clear demonstration that the frequently reported IM near cloud top and HM deeper in cloud can arise as a collective signal from parcels with distinct entrainment–evaporation histories, rather than a true local mixing mode. It offers a compelling reframing of bulk versus local perspectives. The goals and message are clear, and the results have practical value for the community by informing the design and interpretation of in-situ aircraft measurements as well as LES/Lagrangian modeling strategies to diagnose entrainment and mixing in stratocumulus clouds.

I find this a valuable contribution and suitable for prompt publication after revision. I recommend addressing several points of clarification in the discussion and adding a small set of targeted sensitivity tests.

**Comment 1:**

The time evolution of the standard deviation of $\delta q_v$ in Fig. 8 is highly informative. However, as the authors note, post-entrainment descent is the more realistic pathway in marine stratocumulus. I therefore suggest presenting the same diagnostics for a descending (non-isobaric) configuration (e.g., the Control experiment) to assess how adiabatic warming during descent modifies both the characteristic reaction time and the HM IM transition. This would further help LES and Lagrangian trajectory studies that have adopted fixed-lag windows for mixing diagnostics (e.g., Lim & Hoffmann, 2023, 2024), in non-isobaric conditions.

**Comment 2:**

The authors use a monodisperse NaCl aerosol initially and CCN-free entrained air. While this isolates sampling effects, many studies show that entrained aerosols and the pre-mixing droplet size distribution (DSD) shape strongly govern the relative changes in $N$ and $r$, and thus the HM/IM diagnostics (Krueger et al., 2008; Luo et al., 2022; Lim & Hoffmann, 2023). In particular, broader spectra with many small droplets can favor $N$ reductions via complete evaporation, altering the $n$–$r^3$ change. Please either add sensitivity results or, if out of scope, expand the discussion to explain the expected impacts and identify this as a priority for follow-up.

**Comment 3:**

Parcels in real clouds often dwell near the cloud top for a short time after entrainment before descending. Please add a dwell-then-descent variant in which the post-entrainment velocity is held at $w = 0$ for a prescribed $\tau_{\mathrm{dwell}}$ (tens of seconds) and then switched to the descending value used in Control. It would be useful to clarify whether this pathway yields a stronger HM or IM signal

in both the local and bulk perspectives. Moreover, the local HM to IM interpretation may partly reflect insufficient time for droplets to respond, where mixing diagnostics require adequate time for both scalar mixing and microphysical adjustment. The current mixing-diagram framework should explicitly acknowledge this timescale dependence. I therefore recommend adding a short discussion on how analysis-window length and parcel dwell affect the local perspective of the mixing process.

**Grammar and Typo**

- It seems like the unit of entrained air water vapor is wrong. $8.6*10^{-3}$ g / kg seems to be too small.

- In Line 17, the IH characteristic should be the IM characteristic.

- Sometimes, $\psi$ and $\varphi$ are mixed when referring to the homogeneous mixing degree (e.g., Fig. 5). Please fix this for consistency.

**References**

Krueger, S. K., Schlueter, H., & Lehr, P. (2008). Fine-scale modeling of entrainment and mixing of cloudy and clear air. In *15th international conference on clouds and precipitation, cancun, mexico.*

Lim, J.-S., & Hoffmann, F. (2023). Between broadening and narrowing: How mixing affects the width of the droplet size distribution. *Journal of Geophysical Research: Atmospheres*, *128*(8), e2022JD037900.

Lim, J.-S., & Hoffmann, F. (2024). Life cycle evolution of mixing in shallow cumulus clouds. *Journal of Geophysical Research: Atmospheres*, *129*(10), e2023JD040393.

Luo, S., Lu, C., Liu, Y., Li, Y., Gao, W., Qiu, Y., … others (2022). Relationships between cloud droplet spectral relative dispersion and entrainment rate and their impacting factors. *Advances in Atmospheric Sciences*, 1–20. doi: 10.1007/s00376-022-1419-5

---

## Author Comment (AC1)

**Reviewer comments on Why Is Height-Dependent Mixing Observed in Stratocumulus?**

This manuscript examines how to interpret homogeneous versus inhomogeneous mixing (HM/IM) signatures in stratocumulus, integrating insights from in-situ observations and modeling with a framework that explicitly resolves inhomogeneous mixing. This manuscript provides a clear demonstration that the frequently reported IM near cloud top and HM deeper in cloud can arise as a collective signal from parcels with distinct entrainment—evaporation histories, rather than a true local mixing mode. It offers a compelling reframing of bulk versus local perspectives. The goals and message are clear, and the results have practical value for the community by informing the design and interpretation of in-situ aircraft measurements as well as LES/Lagrangian modeling strategies to diagnose entrainment and mixing in stratocumulus clouds. I find this a valuable contribution and suitable for prompt publication after revision. I recommend addressing several points of clarification in the discussion and adding a small set of targeted sensitivity tests.

**Response**: We thank the reviewer for the positive evaluation and encouraging remarks on our work. We also appreciate the thoughtful suggestions, which have greatly improved the completeness of the manuscript. Detailed responses to each comment are provided below.

**Comment 1:**

The time evolution of the standard deviation of δqv in Fig. 8 is highly informative. However, as the authors note, post-entrainment descent is the more realistic pathway in marine stratocumulus. I therefore suggest presenting the same diagnostics for a descending (non-isobaric) configuration (e.g., the Control experiment) to assess how adiabatic warming during descent modifies both the characteristic reaction time and the HM IM transition. This would further help LES and Lagrangian trajectory studies that have adopted fixed-lag windows for mixing diagnostics (e.g., Lim & Hoff- mann, 2023, 2024), in non-isobaric conditions.

**Response**: We thank the reviewer for this insightful comment. In the revised manuscript, we have included the time evolution of the standard deviation of  $\delta q_v$  for the *Control* experiment, as shown in Figure 4a. The corresponding discussion has been added to the revised text.

Figure 4a: Normalized standard deviation of water vapor ( $\delta_{qv}$ ) in the parcel after entrainment for the control experiment. The blue, green, purple and yellow line represents the parcel with EF of 0.1, 0.3, 0.5 and 0.7.

Line 339: "...The normalized standard deviation of water vapor is plotted to illustrate the temporal evolution of the mixing process in the Control experiment (Fig. 4a). The standard deviation of water vapor  $(\delta q_v)$  is calculated at each time step within the one-dimensional domain (20 m in length with a 1 mm grid spacing) and normalized by its value at 1 s after entrainment. The evolution of  $\delta q_v$  reflects the characteristic mixing timescale (Tölle and Krueger, 2014). As shown in Fig. 4a,  $\delta q_v$  peaks immediately after entrainment and decreases over time as mixing between entrained and cloudy air proceeds. Parcels with smaller entrainment fractions (EF) exhibit shorter mixing times than those with larger EF; for example, a parcel with EF = 0.1 reaches equilibrium after roughly 20 s, whereas one with EF = 0.7 requires about 100 s to homogenize water vapor within the domain.

Line 348: "...In the Control configuration, the parcel descends immediately after entrainment at a constant velocity of -1 m s-1, allowing elapsed time to be directly related to distance below the cloud top. Accordingly, three representative height levels: 5 m, 50 m, and 200 m below the cloud top, are selected to characterize distinct stages of the mixing process..."

**Comment 2:**

The authors use a monodisperse NaCl aerosol initially and CCN-free entrained air. While this isolates sampling effects, many studies show that entrained aerosols and the pre-mixing droplet size distribution (DSD) shape strongly govern the relative changes in N and r, and thus the HM/IM di- agnostics (Krueger et al., 2008; Luo et al., 2022; Lim & Hoffmann, 2023). In particular, broader spectra with many small droplets can favor N reductions via complete evaporation, altering the n–r3 change. Please either add sensitivity results or, if out of scope, expand the discussion to explain the expected impacts and identify this as a priority for follow-up.

**Response:** We thank the reviewer for the constructive suggestion. In the revised manuscript, we have added a *CCN-Entrained-Air* experiment, in which the entrained air contains dry aerosols from the free atmosphere that can act as CCN. The results show a consistent transition from IM near the cloud top to HM deeper within the cloud, similar to the control case. In addition, a reduction in mean droplet size and an enhanced HM signature are observed under CCN entrainment, consistent with previous studies (Luo et al., 2021; Lim and Hoffmann, 2023). A detailed description of the experimental setup is provided in **Table 1**, and the corresponding results are presented in **Figure 5**.

|                                                         |         | Č                 |                          |                      |                     |
|---------------------------------------------------------|---------|-------------------|--------------------------|----------------------|---------------------|
| Parameter                                               | Control | Dry Entrained Air | Enhanced
Turbulence   | CCN Entrained
Air | Reduced
Velocity |
| Domain Length (m)                                       |         |                   | 20m                      |                      |                     |
| CCN Concentration (cm -3 )                   |         |                   | 80                       |                      |                     |
| Cloud Top Height (m)                                    |         |                   | 950                      |                      |                     |
| Aerosol Size Distribution                               |         |                   | Monodisperse             |                      |                     |
| Initial solute mass (kg)                                |         |                   | 0.1122*10 -17 |                      |                     |
| Initial aerosol radius (m)                              |         |                   | 0.216*10-6               |                      |                     |
| Type of aerosol                                         |         |                   | NaCl                     |                      |                     |
| Eddy Dissipation Rate (m 2 s -3 ) | 0.0025  | 0.0025            | 0.01                     | 0.0025               | 0.0025              |
| Entrained air temperature (K)                           | 285.77  | 288               | 285.77                   | 285.77               | 285.77              |
| Entrained air water vapor (g/kg)                        | 8.6     | 7.9               | 8.6                      | 8.6                  | 8.6                 |
| Entrained CCN in the dry air                            | N       | N                 | N                        | Y                    | N                   |
| Vertical Air Velocity (ms -1 )               | ±1      | ±1                | ±1                       | ± 1           | ±0.5                |

 $Table 1-Model\ Configuration$

Table 1: Model configurations for the control, dry and turbulent simulation experiment.

Figure 5: Same as Figure 4, but for the four sensitivity experiments: (a)–(b) correspond to the Dry-Entrained-Air experiment; (c)–(d) to the Enhanced-Turbulence experiment; (e)–(f) to the CCN-Entrained-Air experiment; and (g)–(h) to the Reduced-Velocity experiment.

Line 151: "...In addition to the control case, four sensitivity simulations were conducted to evaluate the robustness of the experimental design. The Dry Entrained Air experiment represents the scenario in which the entrained air is drier. Specifically, the model setup is the same as the control one except the entrained air property is estimated using the parcel at 20 m above cloud top experiencing adiabatic descent to cloud top. The selection of the distance of the entrained parcel from cloud top is arbitrary and does not affect the conclusions of this study. The Enhanced Turbulence experiment simulates strongly turbulent environment with EDR set to 0.01 m2 s-3. The CCN entrained Air experiment allows the entrained air containing dry aerosols entrained from free atmosphere. The properties and concentrations of the entrained aerosols are identical to those initially specified within the parcel. ..."

Line 406: "...In the CCN-Entrained-Air experiment (Fig. 5e, f), the normalized  $r^3$  values for each normalized number concentration are smaller than those in the control case, indicating a more pronounced reduction in droplet size. This feature reflects a stronger HM tendency under CCN entrainment, consistent with previous findings that activation of entrained CCN broadens the droplet size distribution toward smaller droplets and amplifies the characteristics of homogeneous mixing (Lim and Hoffmann, 2023; Luo et al., 2022). ..."

Line 422: "...Overall, despite variations in the thermodynamic and dynamic properties of the entrained air, all simulations consistently exhibit an IM signature near the cloud top and a transition toward HM within the cloud, with an increasing degree of HM deeper into the cloud layer. These model-based results align well with aircraft observations in stratocumulus clouds (Yum et al., 2015; Yeom et al., 2021), providing a robust basis for the more detailed analysis presented in the following section..."

**Comment 3:**

Parcels in real clouds often dwell near the cloud top for a short time after entrainment before descending. Please add a dwell-then-descent variant in which the post-entrainment velocity is held at  $\mathbf{w} = \mathbf{0}$  for a prescribed Tdwell (tens of seconds) and then switched to the descending value used in Control. It would be useful to clarify whether this pathway yields a stronger HM or IM signal in both the local and bulk perspectives. Moreover, the local HM to IM interpretation may partly reflect insufficient time for droplets to respond, where mixing diagnostics require adequate time for both scalar mixing and microphysical adjustment. The current mixing-diagram framework should explicitly acknowledge this timescale dependence. I therefore recommend adding a short discussion on how analysis-window length and parcel dwell affect the local perspective of the mixing process.

**Response**: We thank the reviewer for this insightful suggestion. Following the recommendation, we implemented a *dwell-then-descent* modification in the *CCN-Entrained-Air* experiment for the reviewer. In this test, parcels remain stationary at the cloud top ( $w = 0 \text{ m s}^{-1}$ ) for 10 s before descending. The corresponding analyses are shown in **Figure 1** below for the reviewer's reference. As illustrated in **Figure 1c**, introducing a dwell period at the cloud top produces a stronger HM signature and a larger reduction in droplet radius, consistent with enhanced evaporation and mixing during the dwell phase. This supports the interpretation that extended residence time near the cloud top allows more complete droplet adjustment and thus strengthens local HM characteristics. Nevertheless, the overall transition from IM near the cloud top to HM deeper within the cloud remains evident, as shown in the log(L)–log( $\tau$  phase) diagrams (Fig. 1d).

**Figure 1 for reviewer (bulk perspective).** Mixing diagrams for (a)–(b) the *CCN-Entrained-Air* experiment and (c)–(d) the same case with a 10 s dwell at the cloud top prior to descent. The dwell period enhances the HM signature near the top but preserves the overall IM-HM transition with cloud.

The parcel-based mixing diagrams (**Figure 2**) show only minor differences between runs with and without the dwell period. Both exhibit an HM tendency near the cloud top, followed by an increasing trend toward IM influence with depth.

**Figure 2 for reviewer (local/parcel-based perspective):** Local mixing diagram for (a) experiment with entrained CCN, and (b) same case with dwell time of 10s at cloud top before descent.

Additionally, as suggested by the reviewer, we have added more discussions on the local perspective of the mixing process in the revised manuscript:

Line 617: "...Finally, it is noted that this study primarily aims to explain the IM—HM transition within cloud as observed from the bulk perspective. We do not attempt to draw conclusions about the local (e.g. parcelbased) mixing state within cloud. The local mixing behavior can vary depending on the model configuration and analysis approach, and it is strongly influenced by the timescale over which droplet properties (i.e. size and number) adjust following entrainment. For instance, in real cloud parcels may briefly dwell near the cloud top before descending, and the inferred local mixing characteristics therefore depend on this residence time. A longer dwell time near cloud top would permit greater vapor—droplet interaction at cloud top, potentially altering the local mixing signature with depth. A detailed investigation of these time-dependent local mixing processes is beyond the scope of this study..."

**Grammar and Typo**

- It seems like the unit of entrained air water vapor is wrong. 8.6\*10-3 g / kg seems to be too small.
- In Line 17, the IH characteristic should be the IM characteristic.
- Sometimes,  $\psi$  and  $\varphi$  are mixed when referring to the homogeneous mixing degree (e.g., Fig. 5). Please fix this for consistency.

**Response**: We thank the reviewer for the careful and detailed comments. All identified issues have been corrected in the revised manuscript.

**References**

- Krueger, S. K., Schlueter, H., & Lehr, P. (2008). Fine-scale modeling of entrainment and mixing of cloudy and clear air. In 15th international conference on clouds and precipitation, cancun, mexico.
- Lim, J.-S., & Hoffmann, F. (2023). Between broadening and narrowing: How mixing affects the width of the droplet size distribution. *Journal of Geophysical Research: Atmospheres*, 128(8), e2022JD037900.
- Lim, J.-S., & Hoffmann, F. (2024). Life cycle evolution of mixing in shallow cumulus clouds. Journal of Geophysical Research: Atmospheres, 129(10), e2023JD040393.
- Luo, S., Lu, C., Liu, Y., Li, Y., Gao, W., Qiu, Y., ... others (2022). Relationships between cloud droplet spectral relative dispersion and entrainment rate and their impacting factors. Advances in Atmospheric Sciences, 1–20. doi: 10.1007/s00376-022-1419-5
- Jeffery, C. A. and Reisner, J. M.: A study of cloud mixing and evolution using PDF methods. Part I: Cloud front propagation and evaporation, Journal of the atmospheric sciences, 63, 2848-2864, 2006.
- Lim, J. S. and Hoffmann, F.: Between broadening and narrowing: How mixing affects the width of the droplet size distribution, Journal of Geophysical Research: Atmospheres, 128, e2022JD037900, 2023.
- Luo, S., Lu, C., Liu, Y., Li, Y., Gao, W., Qiu, Y., Xu, X., Li, J., Zhu, L., and Wang, Y.: Relationships between cloud droplet spectral relative dispersion and entrainment rate and their impacting factors, Advances in Atmospheric Sciences, 39, 2087-2106, 2022.
- Tölle, M. H. and Krueger, S. K.: Effects of entrainment and mixing on droplet size distributions in warm cumulus clouds, Journal of Advances in Modeling Earth Systems, 6, 281-299, 2014.
- Yeom, J. M., Yum, S. S., Shaw, R. A., La, I., Wang, J., Lu, C., Liu, Y., Mei, F., Schmid, B., and Matthews, A.: Vertical variations of cloud microphysical relationships in marine stratocumulus clouds observed during the ACE-ENA campaign, Journal of Geophysical Research: Atmospheres, 126, e2021JD034700, 2021.
- Yum, S. S., Wang, J., Liu, Y., Senum, G., Springston, S., McGraw, R., and Yeom, J. M.: Cloud microphysical relationships and their implication on entrainment and mixing mechanism for the stratocumulus clouds measured during the VOCALS project, Journal of Geophysical Research: Atmospheres, 120, 5047-5069, 2015.

---

## Author Comment (AC2)

**Review of "Why Is Height-Dependent Mixing Observed in Stratocumulus?"**

This manuscript investigates entrainment—mixing processes in stratocumulus using the Explicit Mixing Parcel Model (EMPM). By emulating virtual aircraft measurements, the authors argue that the frequently observed transition from inhomogeneous mixing (IM) near cloud top to homogeneous mixing (HM) within cloud depth is essentially a collective behavior of multiple parcels sampled at the same height, experiencing distinct entrainment—mixing—evaporation histories, rather than reflecting the true local mixing mechanism. The study compares bulk versus local perspectives, introduces isobaric mixing simulations, and provides a discussion on the implications for interpreting both aircraft and LES data. The topic is highly relevant to the cloud physics community, and the work provides some insight into entrainment—mixing interpretation. However, the revisions are needed to clarify assumptions, better explain contradictory perspectives, and explicitly define terms such as "near cloud top." With these improvements, the paper will be a valuable contribution to the literature.

**Response**: We thank the reviewer for recognizing the value of our work to the cloud physics community and for providing constructive suggestions. We have revised the manuscript according to the comments, and these changes have greatly improved its clarity and completeness. Detailed responses to each comment are provided below.

**Major comments:**

- 1. CCN concentration in entrained air: The manuscript assumes that the entrained dry air is CCN-free. This is a strong simplification, since in reality entrained air frequently contains at least some aerosols that can serve as CCN. I strongly encourage the authors to either (a) perform additional sensitivity experiments with non-zero CCN concentration, or (b) explicitly discuss how this assumption may bias their results and conclusions. Without such treatment, the applicability of the EMPM findings to real stratocumulus environments remains limited.
- 2. Descending velocity after entrainment: A uniform descent rate of -1 m s-1 is imposed. However, the actual descent speed is likely to vary with entrainment fraction (EF), local turbulence intensity, and thermodynamic structure. The authors should justify this choice more thoroughly and, ideally, include sensitivity tests with varying descent rates. A clear discussion of this limitation is necessary to describe how robust the reported IM–HM transition is under different dynamical conditions.

**Response to comments 1 and 2:**

We thank the reviewer for these constructive and insightful suggestions. In response, two additional sensitivity experiments have been conducted and incorporated into the revised manuscript:

- (1) a CCN-Entrained-Air experiment, in which the entrained air contains dry aerosols from the free atmosphere that can act as CCN; and
- (2) a Reduced-Velocity experiment, in which the parcel descends more slowly ( $-0.5 \text{ m s}^{-1}$ ) to assess the influence of vertical velocity on the mixing process.

The model configurations for the new sensitivity experiments are summarized in **Table 1**, and the corresponding results are presented in **Figure 5**. Both additional experiments, together with the other sensitivity tests, show a consistent transition from inhomogeneous mixing (IM) near the cloud top to homogeneous mixing (HM) deeper within the cloud, like the control case. However, each new experiment also exhibits distinct features. The inclusion of CCN in the entrained air enhances HM characteristics near the cloud top through the activation of new small droplets and increased vapor competition, while a reduced descent rate also accelerates homogenization near the cloud top by extending the effective mixing–evaporation time. These results further confirm the robustness of the IM–HM transition in stratocumulus simulated by the EMPM model.

Table 1 - Model Configuration

| Parameter                                               | Control | Dry Entrained Air | Enhanced
Turbulence   | CCN Entrained
Air | Reduced
Velocity |
|---------------------------------------------------------|---------|-------------------|--------------------------|----------------------|---------------------|
| Domain Length (m)                                       |         |                   | 20m                      |                      |                     |
| CCN Concentration (cm -3 )                   |         |                   | 80                       |                      |                     |
| Cloud Top Height (m)                                    |         |                   | 950                      |                      |                     |
| Aerosol Size Distribution                               |         |                   | Monodisperse             |                      |                     |
| Initial solute mass (kg)                                |         |                   | 0.1122*10 -17 |                      |                     |
| Initial aerosol radius (m)                              |         |                   | 0.216*10 -6   |                      |                     |
| Type of aerosol                                         |         |                   | NaCl                     |                      |                     |
| Eddy Dissipation Rate (m 2 s -3 ) | 0.0025  | 0.0025            | 0.01                     | 0.0025               | 0.0025              |
| Entrained air temperature (K)                           | 285.77  | 288               | 285.77                   | 285.77               | 285.77              |
| Entrained air water vapor (g/kg)                        | 8.6     | 7.9               | 8.6                      | 8.6                  | 8.6                 |
| Entrained CCN in the dry air                            | N       | N                 | N                        | Υ                    | N                   |
| Vertical Air Velocity (ms -1 )               | ±1      | ±1                | ±1                       | ±1                   | ±0.5                |

Table 1: Model configurations for the control, dry and turbulent simulation experiment.

Figure 5: Same as Figure 4, but for the four sensitivity experiments: (a)–(b) correspond to the Dry-Entrained-Air experiment; (c)–(d) to the Enhanced-Turbulence experiment; (e)–(f) to the CCN-Entrained-Air experiment; and (g)–(h) to the Reduced-Velocity experiment.

Line 151: "...In addition to the control case, four sensitivity simulations were conducted to evaluate the robustness of the experimental design. The Dry Entrained Air experiment represents the scenario in which the entrained air is drier. Specifically, the model setup is the same as the control one except the entrained air property is using the parcel at 20m above cloud top experiencing adiabatic descent to cloud top. The selection of the distance of the entrained parcel from cloud top is arbitrary and it does not affect the conclusions of this study. The Enhanced Turbulence experiment simulates stronger turbulent environment with EDR set to 0.01 m2 s-3. The CCN-Entrained Air experiment allows the entrained air containing dry aerosols entrained from free atmosphere. The properties and concentrations of the entrained aerosols are identical to those initially specified within the parcel. Finally, the Reduced Velocity experiment represents parcels subjected to a smaller vertical velocity than in the control case. A complete summary of the model configurations for these sensitivity experiments is provided in Table 1..."

Line 406: "...In the CCN-Entrained-Air experiment (Fig. 5e, f), the normalized  $r^3$  values for each normalized number concentration are smaller than those in the control case, indicating a more pronounced reduction in droplet size. This feature reflects a stronger HM tendency under CCN entrainment, consistent with previous findings that activation of entrained CCN broadens the droplet size distribution toward smaller droplets and amplifies the characteristics of homogeneous mixing (Lim and Hoffmann, 2023; Luo et al., 2022). In the Reduced-Descent experiment, the mixing diagram (Fig. 5g) shows a stronger HM characteristics at 5 m below cloud top, accompanied by a greater reduction in droplet radius. This arises because the slower descent velocity allows droplets to remain longer near the cloud top compared to the control one, thereby experiencing longer mixing-evaporation time. An interesting feature of this case is that the fitted lines at the two sampled heights (red and blue) are closely aligned, suggesting small evolution of droplet properties with depth from 50 m to 200 m. This behavior indicates that the environment has nearly reached a homogeneous mixing state, as the reduced descent rate effectively extends the available mixing-evaporation time, allowing the system to equilibrate more rapidly toward HM conditions..."

Line 422: "...Overall, despite variations in the thermodynamic and dynamic properties of the entrained air, all simulations consistently exhibit an IM signature near the cloud top and a transition toward HM within the cloud, with an increasing degree of HM deeper into the cloud layer. These model-based results align well with aircraft observations in stratocumulus clouds (Yum et al., 2015; Yeom et al., 2021), providing a robust basis for the more detailed analysis presented in the following section..."

3. The local perspective suggests HM near cloud top transitioning to IM deeper in cloud, whereas the bulk view shows the opposite. This apparent contradiction is central to the study but is not explained clearly enough.

**Response**: We want to thank the suggestion. We have included more discussions on the local versus bulk perspective in the revised manuscript:

Line 538: "...The HM–IM transition observed from the local perspective appears to contradict the mixing behavior suggested by the bulk perspective. We propose that this inconsistency arises from the differing analytical perspectives. The local perspective indicated in Fig. 6 follows the continuous evolution of individual parcel, revealing the "true" mixing processes. While the bulk perspective captures a "snapshot" of an ensemble of parcels, each with distinct entrainment and mixing histories. At cloud top, the entrained

air is configurated to replace the cloudy air and instantaneously reduce the droplet number. Immediately following entrainment, parcels with large EF experience larger reductions of droplet number, while evaporation is not yet active enough to reduce droplet size. Thus, a collection of multiple parcels with different entrainment events generates an IM signature. As the parcel, as simulated within the model domain, descends deeper into the cloud, mixing with dry air continues and evaporation becomes efficient, leading to a reduction in droplet size. As a result, parcels with larger EF experiencing stronger evaporation and this results in a more pronounced decrease in droplet size and number. Consequently, a collection of parcels with different EFs tends to exhibit a HM signature deeper into the cloud..."

Line 617: "...Finally, it is noted that this study primarily aims to explain the IM-HM transition within cloud as observed from the bulk perspective. We do not attempt to draw conclusions about the local (e.g. parcelbased) mixing state within cloud. The local mixing behavior can vary depending on the model configuration and analysis approach, and it is strongly influenced by the timescale over which droplet properties (i.e. size and number) adjust following entrainment. For instance, in real cloud parcels may briefly dwell near the cloud top before descending, and the inferred local mixing characteristics therefore depend on this residence time. A longer dwell time near cloud top would permit greater vapor—droplet interaction at cloud top, potentially altering the local mixing signature with depth. A detailed investigation of these time-dependent local mixing processes is beyond the scope of this study..."

A related issue is that in several analyses (e.g., Figs. 4-6), the term "near cloud top" is used without a quantitative definition. Since the results depend sensitively on how close the sampling is to the inversion, the lack of a clear threshold (e.g., within 5 m or 10 m below cloud top) makes the interpretation appear ambiguous.

**Response:** We want to thank the reviewer's suggestion. We have explicitly defined the "near-cloud-top region" in the revised manuscript.

Line 351: "...In this study, the "near-cloud-top region" is defined as the layer within 10 m below the cloud-top height (950 m)..."

**Minor comments:**

Ensure that Sc is always defined as stratocumulus upon first use and then used consistently.2. Lines 17 and 78: "IH" seems to be a typographical error and should be corrected to 'IM'.

**Response:** Correction has been made, we thank the reviewer for the careful comment.

3. Line 187: What is the accommodation length 2µm? Please explain it.

**Response:** Eq. 2 is adapted from (Jeffery and Reisner, 2006; Eq.3). In the original paper, *a* is described as "an accommodation length introduced for analytic convenience." We have retained this original description in our manuscript.

Line 270: "...a is the accommodation length taken as 2  $\mu m$ , which is introduced for analytic convenience (Jeffery and Reisner, 2006)..."

4. Figures 4–6: Clear annotations (e.g., "IM-like" vs. "HM-like") on the fitted lines might be helpful to understand these diagrams.

Response: The suggested corrections have been added to the figure caption in the revised manuscript.

**Reference**

Jeffery, C. A. and Reisner, J. M.: A study of cloud mixing and evolution using PDF methods. Part I: Cloud front propagation and evaporation, Journal of the atmospheric sciences, 63, 2848-2864, 2006.

Lim, J. S. and Hoffmann, F.: Between broadening and narrowing: How mixing affects the width of the droplet size distribution, Journal of Geophysical Research: Atmospheres, 128, e2022JD037900, 2023.

Luo, S., Lu, C., Liu, Y., Li, Y., Gao, W., Qiu, Y., Xu, X., Li, J., Zhu, L., and Wang, Y.: Relationships between cloud droplet spectral relative dispersion and entrainment rate and their impacting factors, Advances in Atmospheric Sciences, 39, 2087-2106, 2022.

Yeom, J. M., Yum, S. S., Shaw, R. A., La, I., Wang, J., Lu, C., Liu, Y., Mei, F., Schmid, B., and Matthews, A.: Vertical variations of cloud microphysical relationships in marine stratocumulus clouds observed during the ACE-ENA campaign, Journal of Geophysical Research: Atmospheres, 126, e2021JD034700, 2021.

Yum, S. S., Wang, J., Liu, Y., Senum, G., Springston, S., McGraw, R., and Yeom, J. M.: Cloud microphysical relationships and their implication on entrainment and mixing mechanism for the stratocumulus clouds measured during the VOCALS project, Journal of Geophysical Research: Atmospheres, 120, 5047-5069, 2015.